# Flood Risk Assessment to Enable Improved Decision-Making for Climate Change Adaptation Strategies by Central and Local Governments

Insang Yu [ID] and Huicheul Jung *[ID]

Korea Adaptation Center for Climate Change, Korea Environment Institute, Sejong 30121, Korea
* Correspondence: hchjung@kei.re.kr; Tel.: +82-44-415-7813

**Abstract:** This study assessed the flood risk in the Republic of Korea, considering representative concentration pathway (RCP) climate change scenarios, after applying the concept of "risk" as proposed by the Intergovernmental Panel on Climate Change. For the hazard assessment, hazard indicators were constructed utilizing design rainfall standards, which represented the local flood protection capability, as a flood threshold. We constructed high-resolution spatial images from data of buildings, roads, agriculture areas, and the population that have suffered significant flood damage in the Republic of Korea. We also calculated flood exposure levels by analyzing the scales of the targets in low-lying areas. Environmental and anthropogenic conditions that can directly increase or decrease river flooding and urban flooding were set as indicators and utilized as proxy variables. As a result of the risk assessment, we found 43 risk areas in the historical period, accounting for 19% of the total administrative districts, 42 in the projected period under RCP 4.5 (18%), and 51 in the projected period under RCP 8.5 (22%). This study's results can be utilized by the central government to determine flood risk priority areas in various administrative districts and by the local government to select priority areas to install flood reduction facilities.

**Keywords:** flood; risk assessment; climate change; adaptation; decision-making; indicator

## 1. Introduction

Globally, a disaster related to a weather, climate, or water hazard has occurred every day on average over the past 50 years, killing 115 people and causing USD 202 million in losses daily. In the period between 1970 and 2019, there were more than 11,000 reported disasters attributed to climate and water hazards globally, with just over 2 million deaths and USD 3.64 trillion in losses, according to a comprehensive new report from the World Meteorological Organization (WMO) [1]. According to the United Nations Environment Programme (UNEP) Adaptation Gap Report 2016, such increasing impacts will result in increases in global adaptation costs. It has been estimated that by 2030, these costs will amount to between USD 140 billion and USD 300 billion annually and by 2050 to between USD 280 billion and USD 500 billion [2]. Despite the improvements in flood mitigation measures and technological advancements, floods continue to endanger human lives [3]. This is mainly due to the increasing human settlements and economic assets in floodplains, land-use change, and climate crisis [4–6]. According to the statistical Yearbook of Natural Disaster of the Ministry of the Interior and Safety, the damage per cause of natural hazards in the Republic of Korea (hereinafter referred to as "Korea") during the decade from 2010 to 2019 was as follows: USD 1616 million (53.8%) due to typhoons, USD 1057 million (35.2%) due to heavy rains, and USD 194 million (6.5%) due to heavy snow; such causes account for the majority of damage [7]. Typhoons and heavy rains, which cause floods, accounted for 89% of the total damage, making Republic of Korea particularly vulnerable to flooding among various natural hazards. Damage from flooding is expected to increase in the future depending on adaptation efforts. According to the sixth

assessment report of the Intergovernmental Panel on Climate Change (IPCC), projected increases in direct flood damages are higher by 1.4 to 2 times at 2 °C and 2.5 to 3.9 times at 3 °C compared to 1.5 °C global warming without adaptation. At global warming of 4 °C, approximately 10% of the global land area is projected to face an increase in both extreme high and low river flows in the same location, with implications for planning for all water use sectors. Challenges for water management will be exacerbated in the near, mid and long terms, depending on the magnitude, rate, and regional details of future climate change, and will be particularly challenging for regions with constrained resources for water management [8].

To reduce the risks of natural hazards such as flood, heatwaves, droughts, and landslides, the central and local governments of Republic of Korea have been enacting climate change adaptation strategies every five years. It is critical to decide the priority ranks of risk regions requiring climate change adaptation measures through risk assessment. Scholars with various scientific backgrounds tend to have different understandings of the assessing methodology of risk and vulnerability [9,10]. Social scientists often focus on the community's ability to anticipate, respond to, and recover from risk [11], while engineers and natural scientists sometimes assess risk by estimating damage and loss to elements based on the results of physical impact assessments through modeling [12]. Methodologies of assessing risk are not common and differ depending on the researcher and the purpose of the given research [13,14]. Therefore, the risk assessment method and spatial resolution are typically determined by the agent who establishes climate change adaptation measures and the spatial scope of risk assessment.

As the central government provides financial support for local governments to establish climate change adaptation measures, a risk assessment on the spatial scale of administrative districts is required for the entire country. Since there are temporal and economic limitations in performing a high-resolution risk assessment for a wide range of the entire country, an index-based risk assessment using proxy variables of administrative districts is useful [15–22]. Since local governments plan structural and non-structural adaptation measures and carry out the tasks of implementing plans for risk areas, a high-resolution risk assessment method is appropriate for administrative districts under management [23–29]. In particular, since floods are affected not only by rainfall but also by topography, it is necessary to utilize the results of a high-resolution physical flood simulation considering regional features.

As an indicator-based risk assessment research case, the European Commission evaluates the risks of earthquakes, floods, and tsunamis every two years for countries around the world through the Index for Risk Management (INFORM), and prepares and publishes a report of the same. The INFORM index is a method to simplify a large amount of information about crisis risk so that it can be easily used for decision-making [30]. The German Agency for International Cooperation (Deutsche Gesellschaft für Internationale Zusammenarbeit, GIZ) and European Academy of Bozen/Bolzano (EURAC) jointly commissioned *The Vulnerability Sourcebook*, a comprehensive tool designed to aid in conducting regular vulnerability assessments. They provide practical guidance on how to apply *The Vulnerability Sourcebook*'s approach using the AR5 [31] risk concept using proxy variables [32]. The Global Climate Risk Index (CRI), developed by Germanwatch, analyzes quantified impacts of extreme weather events—both in terms of the fatalities as well as the economic losses that occurred according to world countries [33]. In Republic of Korea, a web-based Vulnerability Assessment Tool to Build Climate Change Adaptation Plan (VESTAP), which has been developed by the Ministry of Environment and the Korea Adaptation Center for Climate Change to assist the central and local governments to establish climate change adaptation measures, is utilized to evaluate vulnerabilities in each administrative district.

The different dimensions of risk such as physical (structural), social, economic, and institutional risk, although possibly differently defined, are connected to each other [9,14,28,29,34]. While central and local governments need a risk assessment method and spatial resolution suitable for each purpose, the application of a common risk assessment method is required

to establish interconnected adaptation plans. This study proposes a flood risk assessment method designed to enable central and local governments' decision-making for the establishment of climate change adaptation measures, and specifically suggests measures which are suited to the prevailing conditions in Republic of Korea. The flood risk in each administrative district unit was evaluated using proxy variables as indicators. The variables were calculated using high-resolution spatial data such as physical flood modeling results, point-based buildings, and grid-based population data. As for the temporal range for the risk assessment, the historical period was determined as 2001–2020 and the projected period as 2021–2040. In terms of climate change scenarios, the following scale was utilized: from RCP 4.5, a positive scenario that can be realistically achieved, to RCP 8.5, the most negative scenario to prepare for the worst case. The representative concentration pathways (RCPs), which are used for making projections, describe four different 21st century pathways of GHG emissions and atmospheric concentrations, air pollutant emissions, and land use. The RCPs include a stringent mitigation scenario (RCP2.6), two intermediate scenarios (RCP4.5 and RCP6.0), and one scenario with very high GHG emissions (RCP8.5) [35].

The central government can utilize the flood risk map of the entire country developed through this study to provide financial support for local governments, and determine the priority ranks of administrative districts that are in need of national adaptation measures. Local governments can use high-resolution flood exposure data of administrative districts managed by the governments to identify priority areas for establishing structural and non-structural adaptation measures for flood protection. Since the flood risk map in this study was developed using the proxy variables calculated with high-resolution spatial data, it is expected that the map will be appropriate for central and local governments to utilize, and for establishing interconnected climate change adaptation measures.

## 2. Materials and Methods

### 2.1. Concept of Risk

In the existing scientific literature, there are many different views on how to systematically address disaster risk, reflected in various analytical concepts and models of diverse complexity [30]. Recently, the concept of climate change risk was systematized through the fifth assessment report of Intergovernmental Panel on Climate Change (IPCC). According to type of risk under the IPCC definition, risk is often represented as the probability of occurrence of hazardous events or trends multiplied by the impacts ensuing if these events or trends occur. Risk results from the interaction of vulnerability, exposure, and hazard [31]. Compound risks arise from the interaction of hazards, which can be characterized by single extreme events or multiple coincident or sequential events that interact with exposed systems or sectors [36]. A multiform flood event is defined as occurring when the hazard and/or impact elements from one flood subtype interact with another flood subtype or another hazard [37]. Emergent risk is a risk that arises from the interaction of phenomena in a complex system; for example, the risk caused due to geographic shifts in human population in response to climate change leading to increased vulnerability and exposure of populations in the receiving region [38]. Regarding the type of risk without an IPCC definition, aggregate risk is defined as the accumulation of independent determinants of risk [39], and amplified risk is the substantial enhancement of background risk through a combination or concentration of determinants of risk in time or space [40]. Cascading risk is one event or trend triggering others; interactions can be one way (e.g., domino or contagion effects) but can also have feedbacks; cascading risk is often associated with the vulnerability component of risk, such as critical infrastructure [41–44]. Interdependent risk is defined as complex systems involving interactions and interdependencies that cannot be separated and lead to a range of unforeseeable risks [45] and multi-risk is the whole risk from several hazards, considering possible hazards and vulnerability interactions entailing both multi-hazard and multi-vulnerability perspectives [46]. Systemic risk results from connections between risks (networked risks), where localized initial failure could have disastrous effects and cause, at its most extreme, unbounded damage [47]. As per the

basic conceptual framework of disaster risk formulated by INFORM, risk is the interaction of hazard and exposure, and vulnerability and capacity measures [48,49], and was thus presented as such in these reports [50–52].

This study applied the concept of flood risk, which is defined as a negative result of the system by the interaction of hazard, exposure, and vulnerability, through a review of the existing scientific literature. Hazard is the potential occurrence of a flood event that may cause loss of life, damage, and loss to property and infrastructure, and exposure is the presence of people, buildings, and infrastructure in places and settings that could be adversely affected. Vulnerability is the propensity or predisposition to be adversely affected and it encompasses a variety of concepts and elements including sensitivity or susceptibility to harm and lack of capacity to cope and adapt [53]. Vulnerability represents the two aspects of sensitivity and capacity, whereas capacity covers coping as well as adaptive capacity.

### 2.2. Methods of Assessing Flood Risk

Risk consists of functions of hazard, exposure, and vulnerability as described in Section 2.1. The risk calculation formula is shown in Equation (1), and was derived from *The Risk Supplement to The Vulnerability Sourcebook* of GIZ and EURAC research [32]. The risk (R′) was calculated by multiplying the composite indicators of hazard (H), exposure (E), and vulnerability (V) by weights, and then adding them together. The final risk ($R_i$) was calculated by standardizing the calculated risk through min–max normalization of Equation (2). Composite indices of hazard, exposure, and vulnerability are composed of several individual indicators. Composite indicators were calculated by multiplying individual indicators by weights in Equation (3) and then adding them together. Since the ranges and units of several individual indicator values are different, standardization through Equation (4) was necessary to calculate composite indicators. In the case of a time-dependent study, $R'_i$ and $Hn'_i$ are historical period values and $R'_{max}$, $R'_{min}$, $Hn'_{min}$, and, $Hn'_{max}$ are historical and projection period values for min–max normalization. If $R'_i > R'_{max}$, the normalized indicator $R_i$ would be larger than 1 [54]. The lowest risk or hazard during historical period among 229 administrative districts becomes the min value, and the highest risk or hazard becomes the max value.

$$R' = \frac{H \times W_H + E \times W_E + V \times W_V}{W_H + W_E + W_V} \tag{1}$$

$$R_{i,\,0\ to\ 1} = \frac{R'_i - R'_{min}}{R'_{max} - R'_{min}} \tag{2}$$

where H, E, and V are composite indicators of hazard, exposure, and vulnerability, respectively. $W_H$, $W_E$, and $W_V$ are weights of hazard, exposure, and vulnerability, respectively. $R'_i$ represents the individual risk to be transformed, $R'_{min}$ is the lowest value for $R'_i$, $R'_{max}$ is highest value for $R'_i$, and $R_{i,\,0\ to\ 1}$ is the normalized risk.

$$H = \frac{H1 \times W_{H1} + H2 \times W_{H2} + \cdots + Hn \times W_{Hn}}{\sum_1^n W} \tag{3}$$

$$Hn_{i,\,0\ to\ 1} = \frac{Hn'_i - Hn'_{min}}{Hn'_{max} - Hn'_{min}} \tag{4}$$

where Hn is an individual indicator of hazard component, $W_{Hn}$ is weight assigned to the individual indicator, $Hn'_{min}$ is the lowest value for $Hn'_i$, $Hn'_{max}$ is the highest value for $Hn'_i$, and $Hn_{i,\,0\ to\ 1}$ is the new value, i.e., the normalized individual indicator of hazard.

*The Vulnerability Sourcebook* suggests a five-class system with the most positive conditions represented by the lowest class and the most negative represented by the highest class [32]. The risk calculation results are divided into 5 grades: very low, low, intermediate, high, and very high at intervals of 0.2, as shown in Table 1.

**Table 1.** Classification of flood risk.

| Color | Range of Risk Value | Risk Classes | Severity |
|---|---|---|---|
| | $0.8 \leq R$ | 5 | Very high |
| | $0.6 \leq R < 0.8$ | 4 | High |
| | $0.4 \leq R < 0.6$ | 3 | Intermediate |
| | $0.2 \leq R < 0.4$ | 2 | Low |
| | $R < 0.2$ | 1 | Very low |

*2.3. Introduction to Study Area*

The study area for flood risk assessment was the entirety of Republic of Korea. The land area of Republic of Korea is 106,286 km$^2$, the population is 51,850,000, and the population density is 487 people/km$^2$. The urban area is 17,789 km$^2$, which is 16.7% of the entire national land area; as 91.8% of the total population, 47,597,000 people, reside in urban areas, the exposure of urban areas to flooding is very high [55]. Administrative districts in Republic of Korea consist of 17 metropolitan cities and provinces and 229 local governments, as shown in Figure 1. As the country is geographically located in the mid-latitude temperate climate zone, the four seasons it experiences are distinct from one another. In the winter, it is cold and dry under the influence of the cold and dry continental high pressure, and in the summer, it is hot due to the high temperature and humidity of the North Pacific high pressure. The annual average temperature is between 10 and 15 °C; August records the highest temperature, between 23 and 26 °C, and January records the lowest temperature, between −6 and 3 °C. The annual regional precipitation is between 1200 and 1500 mm in the central region, between 1000 and 1800 mm in the southern region, 1800 mm in some coastal regions, and between 1500 and 1900 mm in the Jeju Island area. As there are heavy precipitation events (between 50 and 60% of annual precipitation) in summer, the country is vulnerable to flooding.

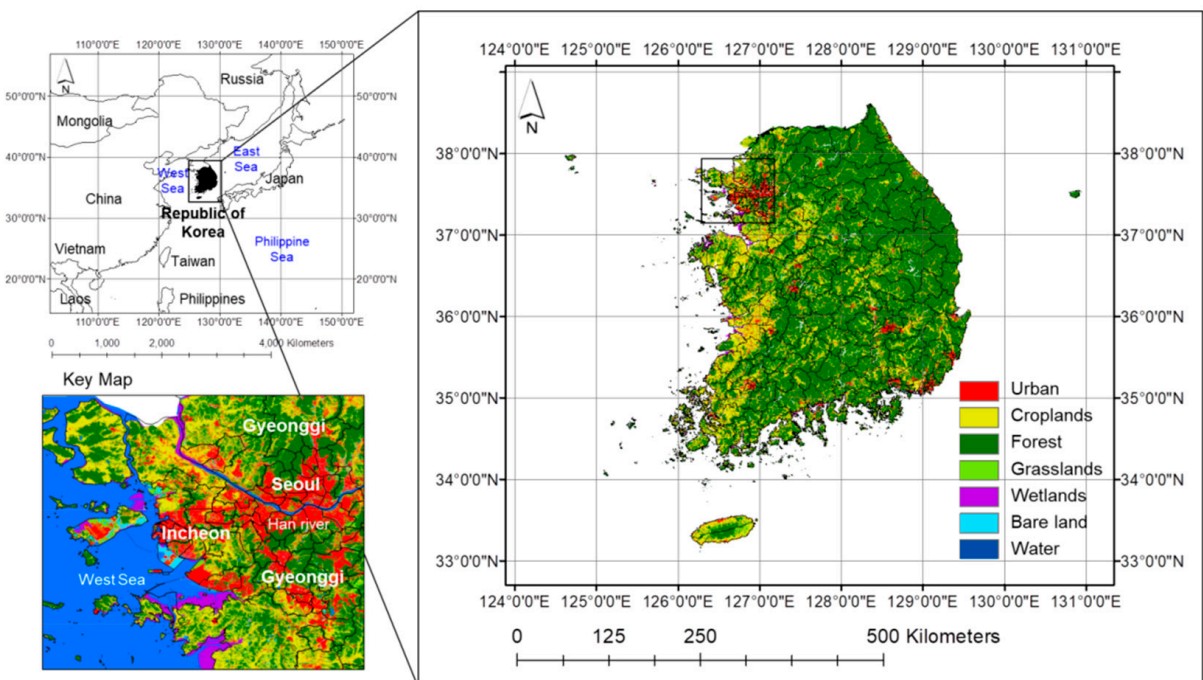

**Figure 1.** Location of study area and its land cover map. The Republic of Korea is a peninsular country located between 33 and 38° latitude and 124 and 132° longitude as shown in the above figure on the left. It is enlarged and shown in the figure on the right. Land cover maps of Seoul city, Incheon city, and Gyeonggi province among administrative districts are enlarged and shown in the lower left figure as an example.

### 2.4. Selecting and Weighting Indicators

Initially, we selected several individual indicators that were highly related to flood risk, hazard, exposure, and vulnerability through literature reviews. The individual indicators for flood risk assessment were selected and improved, as shown in Table 2, through a two-month discussion with seven experts in water resources, flood, and risk assessment with more than 20 years of experience. The discussion dealt with the possibility of constructing a geospatial database (DB), appropriateness, and alternative indices of individual indicators. In previous studies, researchers selected the indicators showing the simple patterns of precipitation, such as the cumulative average maximum 3-day precipitation [56], rainfall [57,58], and the number of precipitation days as hazard indicators. However, even if rainfall is high, flood damage is less likely if flood protection measures are well-designed; even if the precipitation is low, there can be flood damage if the measures are poorly prepared. Therefore, it is necessary to select the indicators that can show how often or how much future rainfall will exceed the design rainfall of the structure for flood protection. In addition, administrative district statistical data such as the population density [56,58,59], the number of households [60], the number of industries, number of commercial units [58], urbanized areas, and green spaces [59], were selected as individual indicators of exposure in previous studies. However, since floods do not occur throughout the administrative districts but in some vulnerable areas, it is necessary to calculate the number of people or buildings in low-lying areas using a high-resolution spatial data analysis. In previous studies, gross regional domestic product (GRDP), ratio of local government financial independence, and unemployment rates [61] were selected as individual indicators of vulnerability. However, these indicators indirectly affect the flood damage without high correlation and have limitations in bringing changes through the climate change adaptation efforts of central and local governments. In this study, as shown in Table 2, individual indicators were selected with direct impacts on flood and risk reduction effects through the establishment of climate change adaptation measures.

**Table 2.** Indicators and their weights for flood risk assessment. Only the weights of the indicators selected for risk calculation are presented in the table. The sum of the indicators' weights and the sum of individual indicator weights in each indicator are 1.

| Composite Indicator | | Individual Indicator | Weight | Period | Source |
|---|---|---|---|---|---|
| Hazard Weight: 0.39 | H1 | Ratio of historical and projected rainfall to design rainfall of river basin (Historical and projected rainfall/design rainfall, dimensionless) Design rainfall:70% of design rainfall corresponding to the river flood warning standard of the Korea Ministry of Environment | 0.34 | Historical: 2001–2020 Projected: 2021–2040 Scenario: RCP4.5/8.5 | KMA, MOE |
| | H2 | Days of historical and projected rainfall in excess of the design rainfall of river basin (historical and projected rainfall > design rainfall, days) | 0.23 | | |
| | H3 | Ratio of historical and projected rainfall to design rainfall of urban watershed (historical and projected rainfall/design rainfall, dimensionless) | 0.15 | | |
| | H4 | Days of historical and projected rainfall in excess of the design rainfall of urban watershed (historical and projected rainfall > design rainfall, days) | 0.18 | | |
| | H5 | Mean annual maximum rainfall of historical and projected period (mm) | 0.10 | | |
| | - | Days of historical and projected rainfall in excess of rainfall of 110 mm (days) | Considered but not selected | | |
| | - | Mean annual maximum 5 days of continuous rainfall of historical and projected period (mm) | | | |

**Table 2.** *Cont.*

| Composite Indicator | | Individual Indicator | Weight | Period | Source |
|---|---|---|---|---|---|
| Exposure Weight: 0.31 | E1 | Area of buildings located on low-lying area (m$^2$) | 0.32 | 2021 | MOLIT |
| | E2 | Area of agriculture located on low-lying area (m$^2$) | 0.12 | 2019 | MOE |
| | E3 | Area of roads located on low-lying area (m$^2$) | 0.18 | 2020 | MOLIT |
| | E4 | The number of people living in low-lying area (people) | 0.39 | 2021 | MOLIT |
| | - | Area of railways located in low-lying area (m$^2$) | | | |
| | - | Population density (people/km$^2$) | | Considered but not selected | |
| | - | The number of disaster-vulnerable people living in low-lying area (people) | | | |
| | - | Area of old (more than 20 years) buildings located in low-lying area (m$^2$) | | | |
| Vulnerability Weight: 0.30 | V1 | Ratio of flooded area in the past 10 years (flooded area/administrative district area, %) | 0.33 | 2019 | MOIS |
| | V2 | Ratio of impervious area (impervious area/administrative district area, %) | 0.19 | 2019 | MOE |
| | V3 | Ratio of built embankment length (built embankment length/planned embankment length, %) | 0.32 | 2020 | KOSTAT |
| | V4 | Ratio of old sewer length (sewer length more than 10 years old/sewer length, %) | 0.17 | 2019 | MOE |
| | - | Gross regional domestic product (GRDP) (USD) | | | |
| | - | Ratio of local government financial independence (%) | | Considered but not selected | |
| | - | Ratio of low-impact development facility area (%) | | | |
| | - | Capacity of drainage pumping station (m$^3$/s) | | | |

KMA: Korea Meteorological Administration, MOE: Ministry of Environment, MOLIT: Ministry of Land, Infrastructure and Transport, MOIS: Ministry of the Interior and Safety, KOSTAT: Statistics Korea.

The indicator weight calculation method can be divided into decision-making and statistical methods. Decision-making methods include the multi-attribute utility theory, analytic hierarch process, fuzzy set principle, and Delphi technique, whereas statistical methods encompass factor analysis, principal component analysis, and the probit model. Among the decision-making and statistical methodologies, decision-making methodologies that derive results based on expert opinions are widely used for calculating the weights of risk indicators and detailed indicators. Among the decision-making methodologies, studies that calculate weights using the analytic hierarchy process (AHP) are mainstream. Since the AHP values the experience of decision-makers, it has the advantage of being able to handle quantitative data as well as qualitative data, which is usually difficult to handle in decision-making, relatively easily. AHP is a universal model that is applicable to various problems requiring decision-making based on simple and clear theories. Therefore, in this study, the AHP, among several decision-making methods, was used to calculate the weight of the selected individual indicators, as shown in Table 2.

*2.5. Definition, Data Acquisition, and Aggregating of Indicators*

The definition of individual indicators selected for flood risk assessment, calculation methods, required data, data construction process and results, and composite indicators for calculating results have been introduced. Composite indicators of hazard, exposure, and vulnerability were calculated by aggregating individual indicators.

2.5.1. Hazard Indicators

Five individual hazard indicators were defined as below. The calculation of the indicators, as shown in Table 3, required historical and projected daily and hourly rainfall, and design rainfall data of river and water basins. Although the spatial resolution of the raw data was different, the estimations of five individual indicators and risks required the conversion of the raw data into the spatial resolution of the 229 administrative districts. The historical daily and hourly rainfall data were collected from 72 stations, and the design rainfall (24 h) data of river basins were obtained from 615 stations; the indicators were calculated with rainfall data of 229 administrative districts, via the area-weighted average method by the Thyssen network. As for the projected daily and hourly rainfall (RCP 4.5 or 8.5) of 1 km spatial resolution by the HadGEM3-RA model constructed and provided by the Korea Meteorological Administration, the spatial average method was applied to calculate administrative district values. Figure 2 indicates the daily rainfall of the historical and projected period under RCP 4.5 and 8.5 scenarios in Gimcheon city, one of 229 administrative districts in Republic of Korea. In the case of Gimcheon city, the daily rainfall exceeding the design rainfall was once in the historical period [Figure 2a], 0 times in the projected period under RCP 4.5, and four times under RCP 8.5 [Figure 2b]; it was found that the excess rainfall was the highest under RCP 8.5. The ratio of design rainfall and severity is the ratio of historical and projected rainfall to design rainfall of river basins and urban watersheds (Figure 2). The frequency refers to the days of historical and projected rainfall in excess of the design rainfall of river basins and urban watersheds (Figure 2). Figure 3 shows the calculation results of composite indicators of hazard and individual indicators, as per climate change scenario in the historical and projected period.

**Table 3.** Required data for calculating individual indicators of hazard and their characteristics.

| Required Data | Period | Sources | Resolution | Note |
|---|---|---|---|---|
| Historical daily and hourly rainfall | 2001–2020 | KMA | 72 points | Observed rainfall from ground stations [62] |
| Projected daily and hourly rainfall | 2021–2040 | KMA | 1 km | The rainfall with HadGEM3-RA model and RCP4.5/8.5 scenario [63] |
| Design rainfall (24 h) of river basin | 100-year (return period) | MOE | 615 points | MOE calculated and issued design rainfall to design structure for protecting river flood [64] |
| Design rainfall (1 h) of watershed | 30-year (return period) | MOIS | 229 administrative districts | MOIS calculated and issued design rainfall to design facilities for protecting urban flood [65] |

KMA: Korea Meteorological, MOE: Ministry of Environment, MOIS: Ministry of the Interior and Safety.

- The ratio of historical and projected rainfall to design rainfall of river basin and urban watershed (Figure 3b,d) is the indicator presenting the intensity of flood occurrence in the river and urban watersheds, and it was assumed that the larger the ratio, the larger the scale of the flood. The percentage of events with the highest precipitation over 20 years was calculated.
- Days of historical and projected rainfall in excess of the design rainfall of river basin and urban watershed (Figure 3c,e) is the indicator showing the frequency of flooding in river and urban watersheds, assuming that the greater the number of days, the higher the likelihood of flooding. The number of days with precipitation that exceeded the design rainfall over 20 years was incorporated.
- Mean annual maximum rainfall of the historical and projected periods (Figure 3f) is the indicator that does not consider the flood protection levels of rivers and urban watersheds, representing the maximum rainfall in the area. This indicator is valid when the protection structures of rivers and urban watersheds are old or cannot function properly for various reasons. The average annual maximum rainfall over 20 years was calculated.

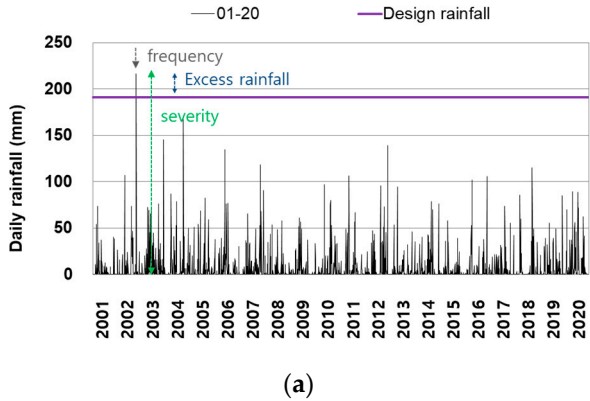

(**a**)

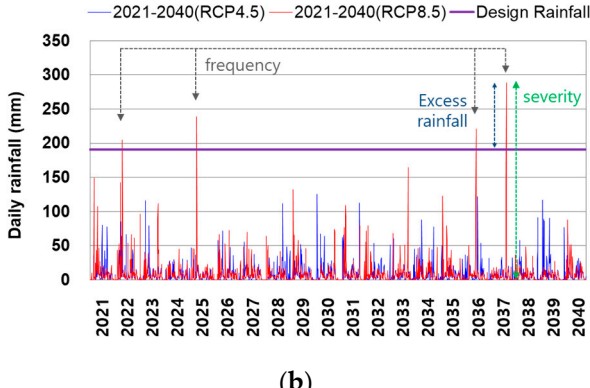

(**b**)

**Figure 2.** Daily rainfall during present and historical period under the RCP 4.5 and 8.5 scenarios of Gimcheon city, one of the 229 administrative districts in Republic of Korea. Since more than 70% of the annual rainfall occurs in the summer, daily rainfall data from June to September were used for every year. (**a**) Historical period (2001–2020). (**b**) Projected period (2021–2040) under RCP 4.5 and 8.5.

2.5.2. Exposure Indicators

In Republic of Korea, the monetary losses incurred due to flood damage over the past decade were found to be KRW 1.874 trillion (87.3%) for public facilities, KRW 63.3 billion for buildings (5.1%), KRW 50.1 billion for agricultural land (4.0%), KRW 43.3 billion for others (3.5%), and KRW 600 million for ships (0.1%). The value of damage to public facilities in 2019 was KRW 30.6 billion (20.4%) for roads, KRW 27 billion for rivers (18.0%), KRW 24.6 billion for erosion control facilities (16.4%), and KRW 18 billion for small streams (12.0%) [7]. Public facilities, buildings, and farmland with high damage were selected as subject to flooding. Among public facilities and roads, the damaged target facilities were selected rather than flood protection facilities. In addition, the population with a large social ripple effect was included as the damaged target.

Individual indicators of exposure related to buildings, agricultural areas, roads, and populations are defined below; the calculation of the indicators, as shown in Table 4, required GIS-based spatial information of the low-lying area, building area, agriculture area, road area, and population data. The selection of low-lying areas required the determination of different reference heights for each watershed. According to the *Cambridge Dictionary*, a low-lying area means land is at, near, or below sea level (*Cambridge Dictionary* 2021). Since this study deals with river flooding caused by heavy rain, not coastal flooding due to sea level rises, the flood level in the 100-year frequency for river design was determined as the reference height for the low-lying area selection. Areas lower than the flood level are vulnerable not only to river floods, but also to inland flooding because rainfall in urban areas is not smoothly discharged. The building area, agriculture area, road area, and population in the low-lying areas were calculated through the spatial data analysis, and composite indicators for each administrative district and individual indicators of exposure were indicated in maps, as shown in Figure 4.

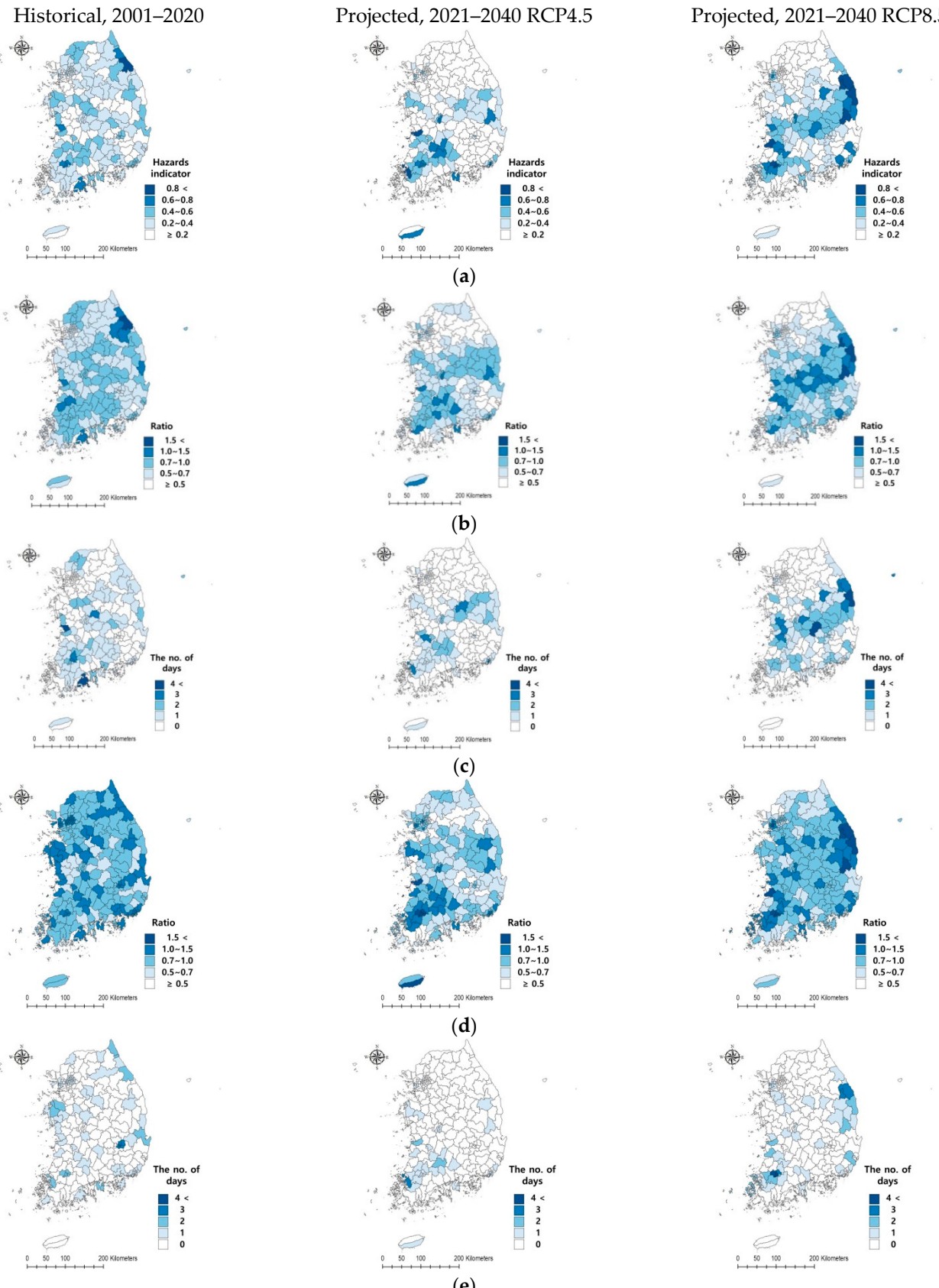

**Figure 3.** *Cont.*

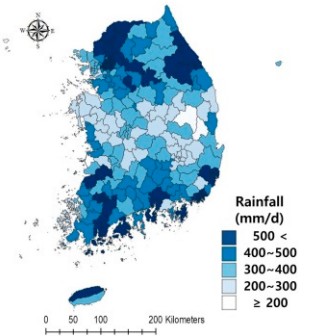
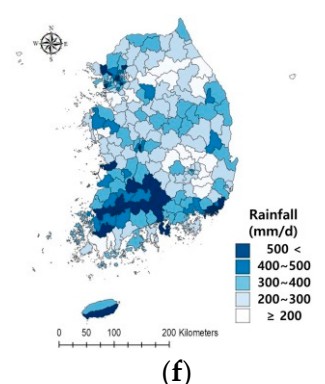
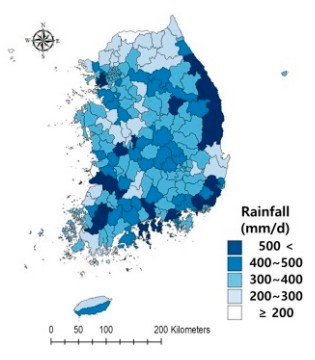

(**f**)

**Figure 3.** Mapping individual and composite indicators of hazard in historical and projected period with HadGEM3-RA climate model and RCP4.5/8.5 climate scenario. Composite indicators were calculated by aggregating five indicators with weights. (**a**) Composite indicators of hazard. (**b**) Ratio of historical and projected rainfall to design rainfall of river basin. (**c**) Days of historical and projected rainfall in excess of the design rainfall of river basin. (**d**) Ratio of historical and projected rainfall to design rainfall of urban watershed. (**e**) Days of historical and projected rainfall in excess of the design rainfall of urban watershed. (**f**) Mean annual maximum rainfall of historical and projected period.

**Table 4.** Required data for calculating individual indicators of exposure and their characteristics.

| Required Data | Period | Sources | Resolution | Note |
| --- | --- | --- | --- | --- |
| Low-lying area | 100-year (return period) | MOE, MOIS | 30 m | Areas lower than the 100-year flood level of national and local rivers |
| Building area | 2021 | MOLIT | Polygon | National Spatial Data Infrastructure Portal [66] |
| Agriculture area | 2019 | MOE | 30 m | Environment Geographic Information Service [67] |
| Road area | 2020 | MOLIT | Polygon | National Spatial Data Infrastructure Portal [66] |
| Population | 2021 | MOLIT | 100 m | National Geographic Information Institute [68] |

MOE: Ministry of Environment, MOIS: Ministry of the Interior and Safety, MOLIT: Ministry of Land, Infrastructure and Transport.

- "Low-lying area" was defined as an area lower than the flood level in the 100-year frequency, which is the return period for the design of flood protection structures of national and local rivers in Republic of Korea.
- Areas of buildings, agriculture, roads, and the populations in low-lying areas are individual indicators of the total buildings, agricultural land, roads, and populations that are expected to be exposed to flooding because they are located below the river flood level. It is assumed that the larger the value, the greater the possibility and magnitude of exposure.

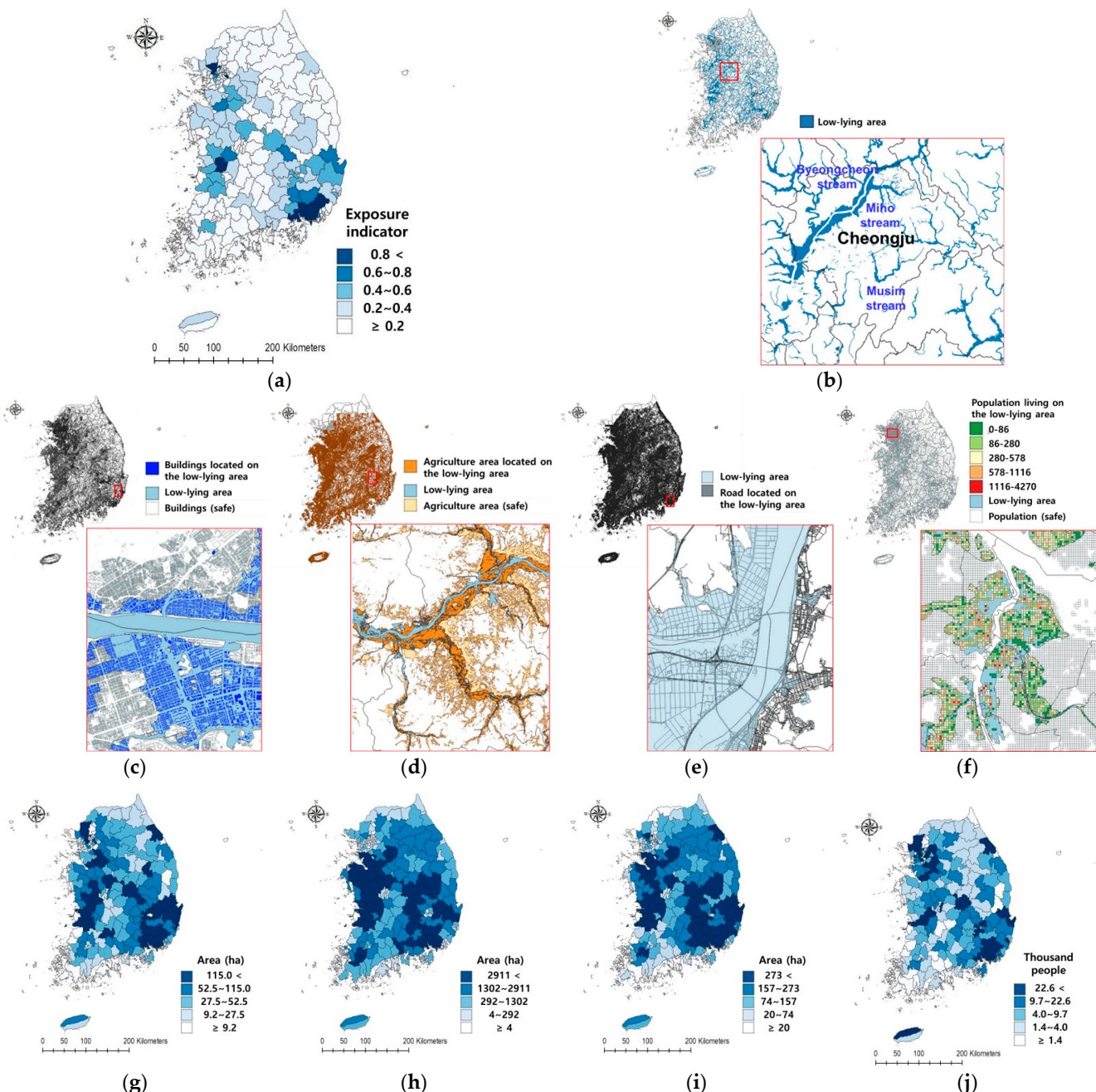

**Figure 4.** Mapping individual and composite indicators of exposure. Individual indicators were calculated by converting grid and polygon types of building, agriculture, road area, and population spatial information into the resolution of administrative district values. Composite indicators were calculated by aggregating four indicators with weights. (**a**) Composite indicators of exposure. (**b**) Low-lying area. (**c**) Buildings located in the low-lying area. (**d**) Agriculture areas located in the low-lying area. (**e**) Roads located oi the low-lying area. (**f**) Population located in the low-lying area. (**g**) Building area. (**h**) Agriculture area. (**i**) Road area. (**j**) Population.

2.5.3. Vulnerability Indicators

Four individual indicators of vulnerability have been defined below; the calculation of the indicators, as indicated in Table 5, required data on areas flooded in the past 10 years, impervious areas, built embankment lengths, and old sewer length data, and they have been mapped as shown in Figure 5. As the flooded area and impervious area data was provided in polygon and 30 m raster formats, respectively, the individual indicators in

each administrative district were calculated through a spatial data analysis. Built embankment length and old sewer length data were provided by the national administrative district statistics.

**Table 5.** Required data for calculating individual indicators of vulnerability and their characteristics.

| Required Data | Period | Sources | Resolution | Note |
|---|---|---|---|---|
| Area flooded in the past 10 years | 2010–2019 | MOIS | Polygon | A map developed by MOIS by surveying areas that have been flooded in the past 10 years |
| Impervious area | 2019 | MOE | 30 m | Environment Geographic Information Service [67] |
| Built embankment length | 2020 | MOE | 229 administrative districts | Korean statistical information service [69] |
| Old sewer length | 2019 | MOE | 229 administrative districts | Statistics of sewerage [70] Sewer length more than 10 years old |

MOE: Ministry of Environment, MOIS: Ministry of the Interior and Safety.

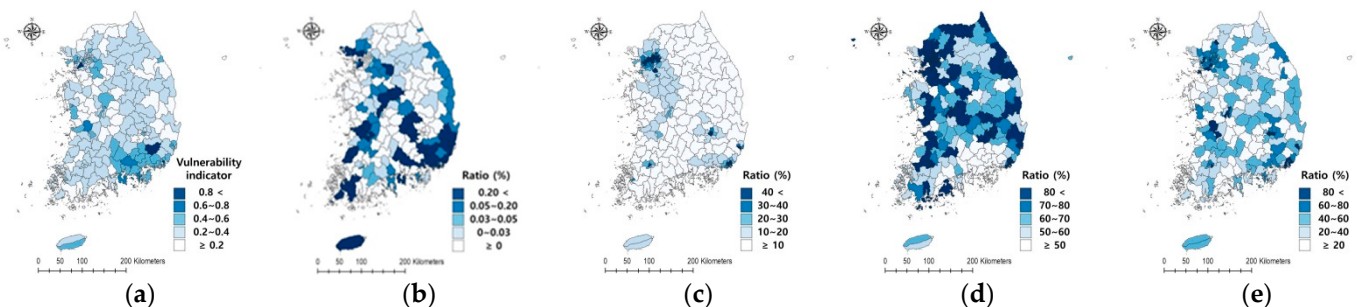

**Figure 5.** Mapping individual and composite indicators of vulnerability. (**b**,**c**) These areas were calculated by converting polygon and grid types of flooded and impervious areas from land cover maps into the resolution of administrative district values; (**d**,**e**) are statistics of administrative statistics. Composite indicators (**a**) were calculated by aggregating four indicators with weights. (**a**) Composite indicators of vulnerability. (**b**) Ratio of area flooded in the past 10 years. (**c**) Ratio of impervious area. (**d**) Ratio of built embankment length. (**e**) Ratio of old sewer length.

- The ratio of the area flooded in the past 10 years is an index indicating the ratio of flooded areas that have occurred in the past decade. The higher ratio was considered as areas with frequent flood damage, assuming that there would be more flood damage.
- The ratio of impervious area is an index indicating the ratio of the impervious area; since the higher ratio indicates lower soil infiltration of rainfall and higher surface runoff, it was assumed that the higher flood peak aggravated the flood damage.
- The ratio of built embankment length is an index indicating the ratio between the built embankment length and the river length necessary for embankment installation. It was assumed that the higher ratio indicated the lower possibility of river flooding.
- Ratio of old sewer length is an index indicating the ratio of the length of old sewer pipelines. It was assumed that the higher ratio referred to the higher probability of inland flooding, due to poor rainfall discharge in urban areas.

## 3. Results

To present the scientific basis for establishing the climate change adaptation measures of central and local governments for flood protection, the following analyses must be considered: (1) risk area screening by administrative districts through indicator-based flood risk assessment in Republic of Korea, (2) cause analysis of flood risks through the analysis of composite and individual indicators for administrative districts with high risks,

and (3) the high-resolution spatial analysis of administrative districts with high risks. The outcomes can be found in Results and Discussion.

### 3.1. Analysis of Spatiotemporal Changes in Flood Risk of Republic of Korea according to Administrative District

The flood risks of the historical period 2001–2020 and projected period 2021–2040 under RCPs 4.5 and 8.5 were calculated through weighted individual and composite indicators, and the temporal and spatial changes are shown in Figure 6. In order to analyze the change of risk areas in line with the temporal changes, areas with high severity of risk or higher were defined as risk areas. There were 43 risk areas in the historical period, accounting for 19% of the total administrative districts; it was analyzed that there were 42 in the projected period under RCP 4.5 (18%) and that there were 51 in the projected period under RCP 8.5 (22%). The numbers of risk areas in the historical and projected periods under the RCP 4.5 scenario were similar to each other, but it was found that the projected period under RCP 8.5 indicated somewhat more risk areas, compared to the historical period. As a result of the spatial change analysis of the risk areas, it was predicted that the risks in the northern region in the historical period would gradually decrease, and the risks in the central and southern regions in the projected period would increase.

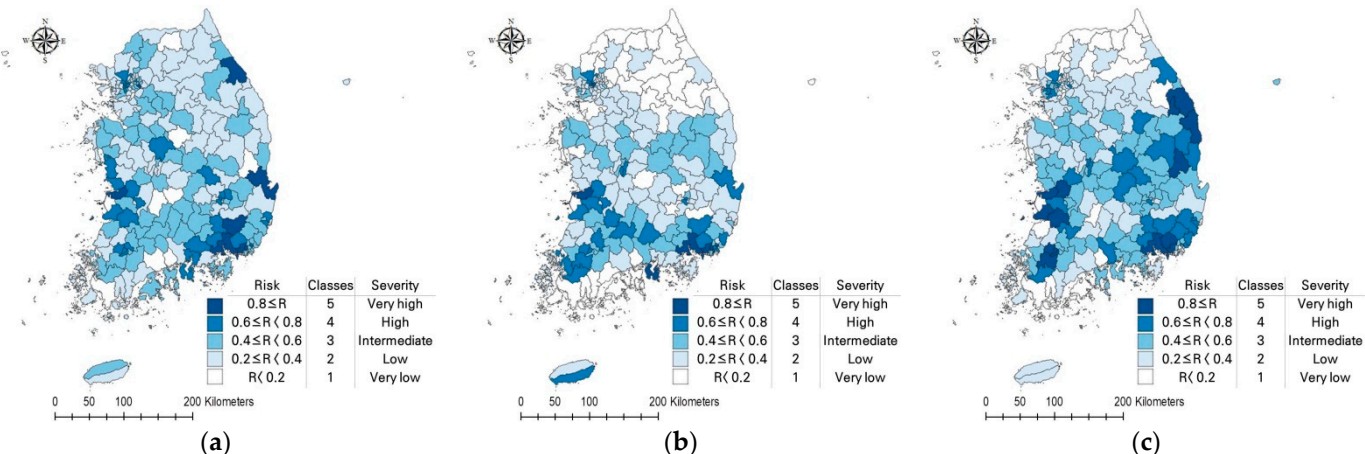

**Figure 6.** Changes in spatiotemporal flood risk of historical and projected period under the RCP4.5 and 8.5 scenarios according to administrative district. (**a**) Historical (2001–2020). (**b**) Projected (2021–2040 under RCP4.5). (**c**) Projected (2021–2040 under RCP8.5).

### 3.2. Analysis of Flood Causes in Very High-Risk Administrative Districts through Composite and Individual Indicators

The flood risk in the historical and projected periods was evaluated by administrative district, and the composite indicators of risk, such as hazard, exposure, and vulnerability, were analyzed. Figure 7 indicates that the risk and composite indicators of the region had the severity rating of "very high", which is the highest priority in establishing climate change adaptation measures among the five grades of severity. Administrative districts were arranged in ascending order based on risk under the RCP 8.5 scenario according to the most severe flood risk. To analyze the cause of the high risk of flooding, the indicator with the highest value among the three composite indicators is displayed in a navy blue shade. It was found that the hazard composite indicator had a great influence on higher flood risks in 10 out of 16 regions, exposure in four regions, vulnerability in one region, and exposure and vulnerability in one region.

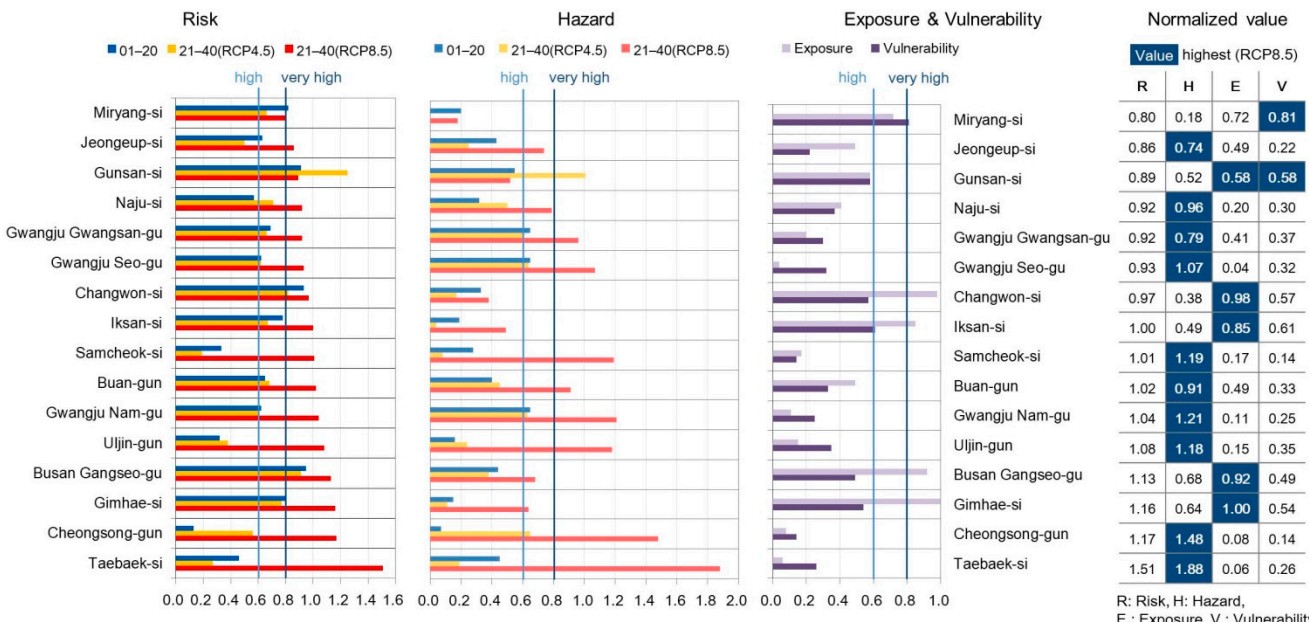

**Figure 7.** Flood risk and composite indicators of very high-risk areas. Administrative districts are sorted in ascending order based on the flood risk of projected period (2021–2040) under the RCP 8.5 scenario.

As a result of the analysis of individual indicators of 16 administrative districts with a risk rated as "very high", the event exceeding the design rainfall in the projected period under the RCP 8.5 scenario was equal to an average of 1.9 times and a maximum of 4 times in the river and urban area; the ratios of maximum rainfall to design rainfall were calculated as an average of 1.3 and a maximum of 2.5 in the river basin, and an average of 1.6 and a maximum of 3.0 in the urban area. The average maximum daily rainfall was 427 mm, and the maximum was 883 mm. The events exceeding the average design rainfall in Republic of Korea were 0.67 and 0.34 in the river and urban areas, respectively; the ratios of maximum rainfall and design rainfall were 0.72 and 0.89, respectively, and the average daily rainfall was 256 mm. The average building area in the low-lying areas was 155 ha and the maximum was 430 ha; the average road area was 410 ha and the maximum was 790 ha; the average agricultural area was 5358 ha and the maximum was 15,412 ha; the average population was 25,097 and the maximum was 84,962. Meanwhile, the average exposure scales in Republic of Korea were 64 ha, 162 ha, 1786 ha, and 19,289 (people), respectively. The average ratio of flooded areas in the past 10 years was 0.4% and the maximum was 3.7%; the average ratio of the impervious areas was 14.1% and the maximum was 41.8%; the average ratio of built embankment length was 66% and the maximum was 100%; the average ratio of old sewer length was 48.6% and the maximum was 80.8%. Meanwhile, the average in Republic of Korea was 0.3%, 20.5%, 76.5%, and 43.9%, respectively.

The administrative districts with the highest values of hazard, exposure, and vulnerability were Taebaek-si, Gimhae-si, and Miryang-si; the individual indicators were analyzed in more detail for these regions, as illustrated in Figure 8. Among five individual indicators of hazard in Taebaek-si, four indicators (H1, H3, H4, H5) were found to have the severity rating of "very high"; one indicator (H2) was found to have the severity rating of "high"; one indicator (V4) out of four individual indicators of vulnerability was found to have the severity rating of "high" (Figure 8). During the projected period under the RCP 8.5 scenario, it was predicted that there would be three events (H2) exceeding 70% of Taebaek-si's design rainfall in the river basin (367 mm); of which the rainfall of the most severe event was 883 mm (H5), which was predicted to be 2.4 times (H1) of the design rainfall (Table 6). Meanwhile, during the historical period, H2 was calculated once, and H1 0.74 times, and it was found that there was no event exceeding the design rainfall. It was predicted that

an event exceeding 64.6 mm/hr, the design rainfall of the urban watershed, in Taebaek-si, would occur three times (H4) in the projected period under the RCP 8.5 scenario; the maximum rainfall would be 190 mm/hr, which was predicted to be three times (H3) the design rainfall (Table 6). On the other hand, during the historical period, H4 was calculated once and H1 1.17 times; there was an event exceeding the design rainfall in the past as well, but the excess rainfall was not found to be high. As the ratio of 10 year-old sewer length in Taebaek-si was 70.1%, which was higher than the national average of 43.9%, a sewer improvement project in major urban areas was found to be necessary. In Taebaek-si, as the predicted frequency and severity of rainfall events exceeding the design rainfall of river and urban areas during the projected period under the RCP 8.5 scenario were high, the flood risk was estimated to be high.

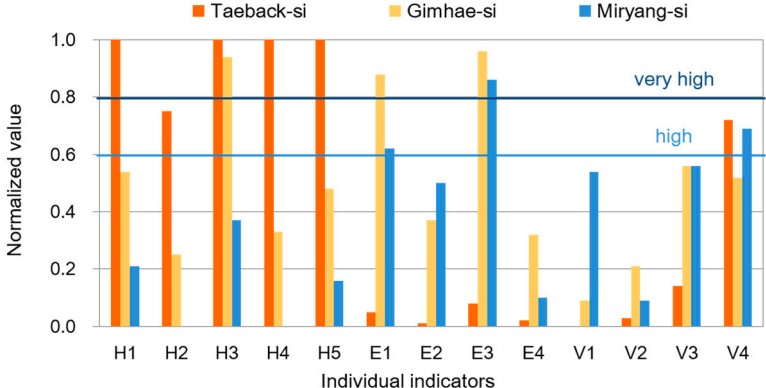

**Figure 8.** Normalized individual indicators of three administrative districts representing very high-risk areas under the RCP 8.5 scenario. Among the three risk components, hazard in Taebaek-si, exposure in Gimhae-si, and vulnerability in Miryang-si have the greatest impact on risk.

**Table 6.** Raw data of individual indicators of three administrative districts representing very high-risk areas under the RCP 8.5 scenario.

| Area | H1 (−) | H2 (No.) | H3 (−) | H4 (No.) | H5 (mm) | E1 (ha) | E2 (ha) | E3 (ha) | E4 (No. of People) | V1 (%) | V2 (%) | V3 (%) | V4 (%) |
|---|---|---|---|---|---|---|---|---|---|---|---|---|---|
| Taebaek-si | 2.4 | 3 | 3.0 | 3 | 883 | 23 | 114 | 63 | 5805 | 0.0 | 3.3 | 87.0 | 70.1 |
| Gimhae-si | 1.1 | 1 | 1.3 | 1 | 436 | 393 | 5705 | 762 | 84,962 | 0.6 | 18.4 | 48.0 | 51.2 |
| Miryang-si | 0.7 | 0 | 0.8 | 0 | 235 | 275 | 7777 | 682 | 25,632 | 3.7 | 8.5 | 48.0 | 67.8 |

One indicator (H3) out of the five individual indicators of hazard in Gimhae-si was found to have the severity rating of "very high"; two indicators (E1 and E3) out of the four individual indicators of exposure were evaluated to have the severity rating of "very high" (Figure 8). During the projected period under the RCP 8.5 scenario, it was predicted that there would be one event (H2) exceeding 70% of the design rainfall of the river basin (386 mm) in Gimhae-si, and the rainfall would be 393 mm (H5), which was predicted to be 1.1 times the design rainfall (H1) (Table 6). An event exceeding the design rainfall of urban watershed (88 mm/hr) was projected to occur once (H4) during the projected period under the RCP 8.5 scenario; the rainfall was predicted to be 112 mm/hr, which was predicted to be 1.3 times (H3) the design rainfall (Table 6). On the other hand, during the historical period, it was found that there was no event exceeding the design rainfall of river and urban areas. The areas of building and road located in the low-lying areas were calculated to be 393 ha and 762 ha, respectively, which was 5 times and 4 times higher than the national average of 68 ha and 162 ha, respectively, indicating that the exposure was estimated to be high. It must be noted that Gimhae-si has concentrated private and public facilities in low-lying areas; even if moderate flooding occurs, the exposure is high and greater damage is expected to occur. The design rainfall of the river is appropriate, but the design rainfall of the urban watershed would be exceeded 1.3 times in the projected period; it is therefore

required that governments establish flood protection measures, such as the reduction of impermeability in urban areas where buildings and roads are concentrated, and improve the old sewer.

All individual indicators of hazard in Miryang-si were analyzed to be below the severity of "moderate"; two individual indicators of exposure (E1 and E3) were evaluated to have the severity rating of "high"; one individual indicator of vulnerability was found to have the severity rating of "high" (Figure 8). During the historical and projected period, as there was no event exceeding the design rainfall of river and urban areas, the current design rainfall was analyzed as appropriate. The areas of building and road located in the low-lying area were 275 and 682 ha, respectively, which were four times higher than the national average of 68 ha and 162 ha, respectively, indicating that the exposure was estimated to be high (E3). Meanwhile, the area of agriculture located in a low-lying area was 7777 ha, indicating the severity rating of "moderate", but it was estimated to be four times higher than the national average of 1786 ha, predicting that agricultural areas are highly likely to be exposed to flooding. The ratio of the 10-year old sewer length was 67.8%, which was 1.5 times higher than the national average of 43.9%, meaning that sewer aging is a serious issue.

## 4. Discussion

### 4.1. Causes of Increased Flood Risk and Suggested Countermeasures for Central Government

Areas where the risk was expected to increase in the projected period compared to the historical period are as shown in Figure 9. As most areas where the risk was expected to increase under the RCP 8.5 scenario [Figure 9b] encompass the areas with higher risks under the RCP 4.5 scenario [Figure 9a], the analysis results were mainly described under the RCP 8.5 scenario in this study. The areas with higher flood risk in the projected period were analyzed to be part of most central, southwestern, and southeastern regions. The central region was expected to have high mean annual maximum rainfall, as shown in Figure 9c,e, and the current design rainfall of the river basin and urban watershed was relatively low, as indicated in Figure 9d,f. In other words, although the central region currently has a relatively low adaptive capacity (design rainfall) of structural and non-structural measures for river and urban flooding protection, a large amount of precipitation was expected in these regions in the projected period. The design rainfall of the river basin and urban watershed was determined by a frequency analysis of the annual maximum rainfall data for the past three decades; it can be seen that these areas did not have much rainfall relative to the past, as shown in Figure 3f. This result shows that for the flood protection structures designed without considering climate change impacts in the past, climate change adaptation and building resilience are possible only when the spatiotemporal changes in future rainfall caused by climate change are reflected in the design. Since the central government's role is to improve the design rainfall for flood protection structures in Republic of Korea, the results of hazard and risk analysis per administrative district are expected to be utilized to support the central government's decision-making to establish climate change adaptation measures.

### 4.2. Determination of Flood Risk Areas through Spatial Information in High-Resolution for Local Government

In order to present the scientific basis for the establishment of a Climate Change Adaptation Plan to reduce flooding by the central and local governments, the selection of high-resolution risk areas is required. High-resolution spatial data have already been used for exposure analysis, and Gimhae-si, which has high levels of hazard, exposure, and vulnerability, was selected as a pilot area, and high-resolution risk areas were selected, as shown in Figure 10. Three selected risk areas with flood-exposed roads, buildings, agriculture areas, and populations are indicated as shown in Figure 10a.

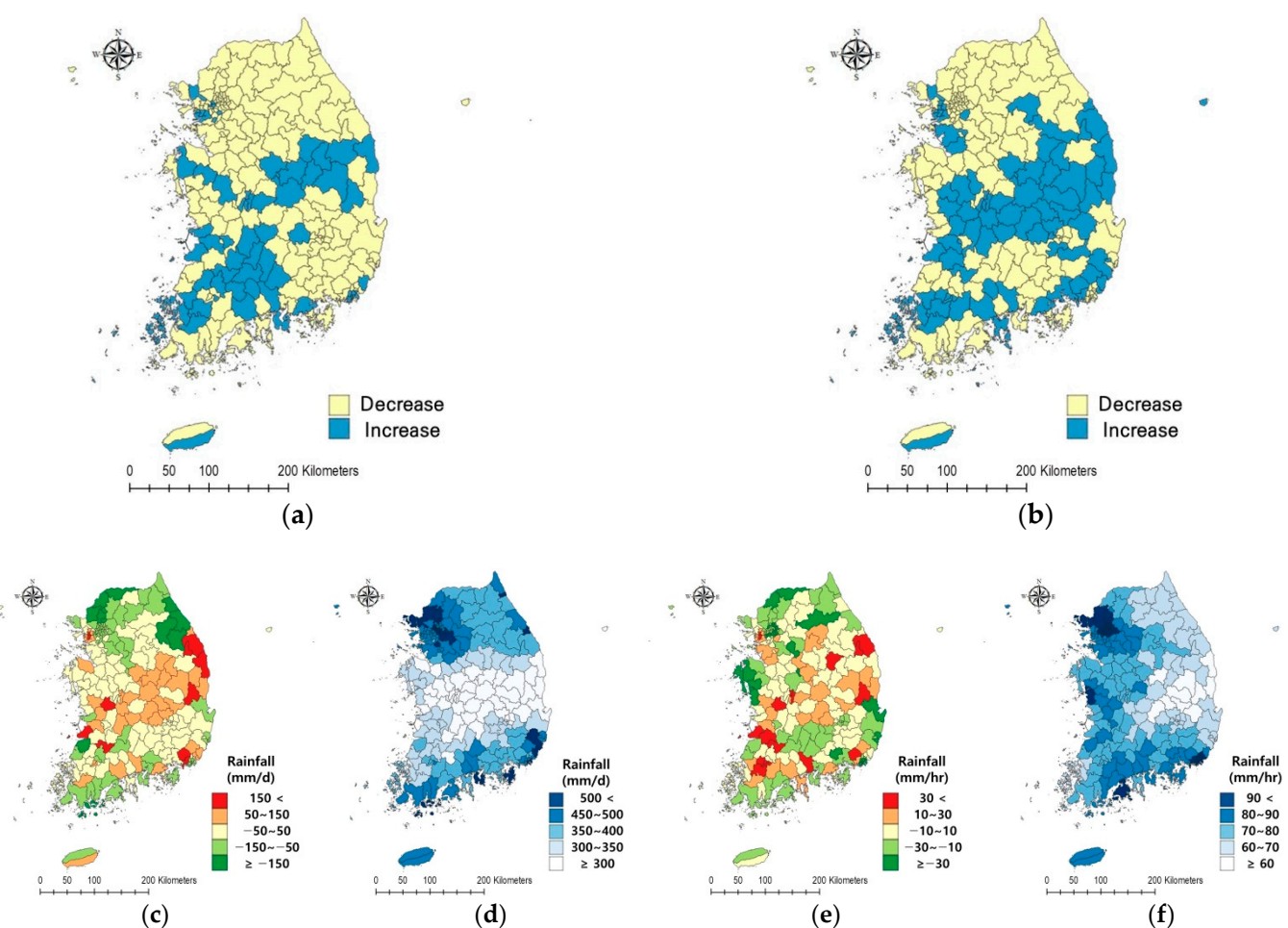

**Figure 9.** Changes in the spatiotemporal flood risk of the historical and projected periods under the RCP 4.5 and 8.5 scenarios according to administrative district. (**a**) Difference of risk between the historical (2001–2020) and projected (2021–2040) periods under RCP4.5. (**b**) Difference of risk between the historical (2001–2020) and projected (2021–2040) periods under RCP8.5. (**c**) Difference of mean annual maximum daily rainfall between historical and projected periods under RCP8.5. (**d**) Design rainfall of river basin. (**e**) Difference of mean annual maximum hourly rainfall between historical and projected periods under the RCP8.5. (**f**) Design rainfall of urban watershed.

The flood risk areas in the center of the north (Figure 10b) are mostly composed of agricultural areas, and some farmers dwell there. Since this area is underdeveloped, there are sufficient spaces near the river to protect against river flooding through the construction and expansion of embankments. In addition, flooding in agricultural areas occurs because the level of rivers rises due to heavy rain, and the rain in agricultural areas is not discharged into rivers. Since this area is already permeable, flood protection measures in the urban areas cannot be effective. It is necessary to install and control drainage gates to prevent the backflow of rivers with high flood levels, and to install drainage pumps to discharge rain that falls on agricultural areas to rivers beyond the embankment.

The areas northwest of Gimhae-si (Figure 10c) are urban areas where buildings and populations are concentrated in low-lying areas near the Nakdong River; it is expected that the areas will experience greater casualties and property damage than other regions, despite a smaller scale of flooding. Since this area has already undergone a lot of urban development near rivers, and is an area with a high impermeability, it is expected that there will be limitations in establishing large-scale structural measures such as more embankments and reservoir construction. Therefore, we expect the following measures to be effective: small-scale structural measures such as drainage pumping station construction

and capacity increase, and reduction of impermeability through the construction of low-impact development facilities, as well as non-structural measures such as preparation of flood evacuation maps and training, construction of a flood warning system, and inducing the purchase of flood insurance.

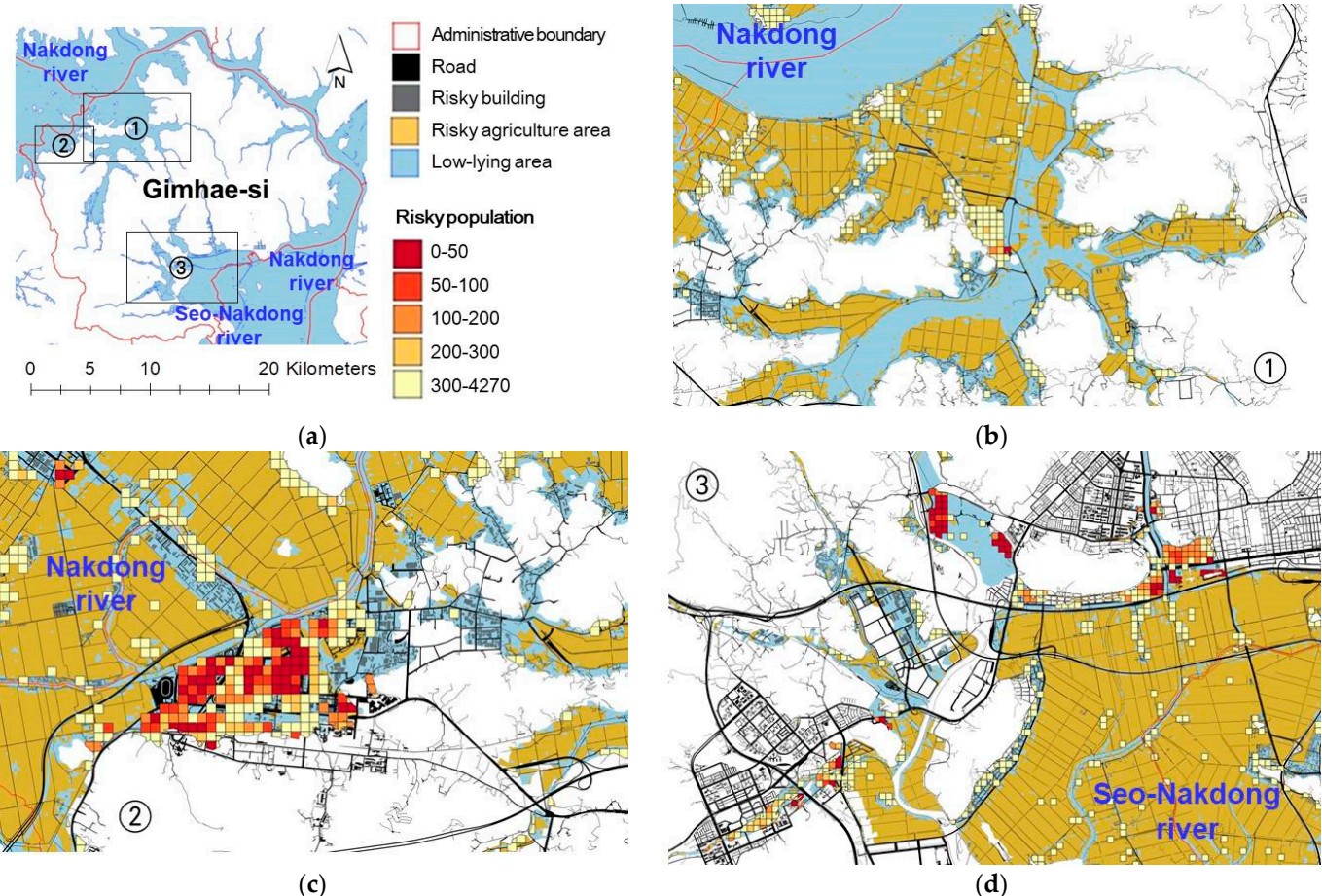

**Figure 10.** Selection of risk areas of Gimhae-si in high-resolution for supporting decision-making of local government. (**a**) Selected risk regions and low-lying areas of Gimhae-si. (**b**) Risk region with dense agriculture area. (**c**) Risk region with dense population and buildings. (**d**) Risk area with multiple exposures.

The flood risk areas in the center of the south (Figure 10d) are mostly composed of agricultural areas, but the areas are complex, since buildings and populations are concentrated in some areas. Therefore, it is necessary to classify the watershed as urbanized or agricultural, and establish localized flood protection measures, as described earlier.

Although this study diagnosed the flood risk in Republic of Korea through proxy indicators, it is necessary to review the adequacy of the design rainfall of river and urban areas, through physical-based rainfall-runoff modeling under the climate change scenario. Physical-based rainfall-runoff modeling on a country scale has temporal and economic limitations. If the modeling is performed on high-risk areas as determined in this study first, it is expected to assist the central and local government to make decisions for establishing efficient climate change adaptation measures and building resilience. The national rivers in Republic of Korea are managed by the central government, and the local rivers and urban watersheds are managed by the local government. The central government needs to review, in detail, the adequacy of flood protection structures, such as the embankment extension and height improvement of national rivers in the administrative districts with high H1 and H2 indicators. The local governments need to evaluate, in a detailed manner, the local rivers in the administrative districts with high H1 and H2 indicators, and the

impermeability, drainage pump capacities, sewage pipe capacity and aging, and capacity of low-impact development facilities in the areas with high H3 and H4 indicators. The priority areas for establishing structural and non-structural flood protection measures can be selected, as shown in Figure 10, by analyzing the scales of the buildings, agriculture areas, roads, and population in the low-lying areas.

### 4.3. Comparison with Previous Studies

Several studies on indicator-based flood risk assessment have been conducted nationally and internationally. The Korea NDMI assessed the index-based flood risk by administrative district using historical meteorological data and social and economic statistics [71]. The Korea Environment Institute evaluated the risks caused by various natural disasters, including floods, and developed policies to establish countermeasures [72]. The research results of these organizations are useful for the central government to select administrative districts that are at risk of flooding, but there are limitations in decision-making for detecting high-resolution risk areas or establishing countermeasures. Since the previous research did not use an index that considers the regional adaptation capacity, such as the design rainfall of the flood protection structures, it is difficult to make a decision for strengthening capacity. In addition, there is a limit in selecting high-resolution risk areas because flood modeling results through numerical simulation were not used. This study contributes to strengthening the decision-making ability of the central and local governments by improving the weakness of the previous research. As a project of risk assessment on a global scale, the European Commission calculates the index for risk management of natural disasters by country and publishes it every two years. Its methodology for estimating risks is very similar to the one used in this study, such as applying the IPCC risk concept and calculating indicators using criteria for issuing forecasts and warnings of disaster. Spatially detailed risk information is not provided since the risk was assessed worldwide using national-level statistical data. There is a limit to directly linking risk assessment results with the establishment of measures, because high-resolution spatial information is essential for selecting risk area. The European Commission scores indicators using absolute criteria rather than relative indicator scoring methods such as z-score and min–max normalization. Developing an absolute standard for scoring indicators is difficult, but it would be a very useful method for periodic risk monitoring. Such methods will be applied in these studies in the future [52].

### 4.4. Future Works

This study used rainfall data under the RCP scenario for flood risk assessment. Recently, global climate data under the SSP (shared socioeconomic pathways) scenario were produced through the CIMIP6 (Coupled Model Intercomparison Project). This study conducted a risk assessment targeting 229 administrative districts with an area of 2.8–1816 km$^2$ to provide scientific support for local governments to establish climate change adaptation measures. Therefore, high-resolution (1 km) climate data are required, but high-resolution climate data under the SSP scenario have not yet been developed in the Republic of Korea, so the climate data of the RCP scenario were used. By downscaling the climate data under the SSP scenario produced through various climate models, high-resolution climate data will be produced and applied to risk assessment in the future.

Building, road, agriculture area, and population data were selected as the target of flood risk assessment, and only the magnitude of the targets located in the low-lying areas was considered. However, the condition of the target is also very important in risk analysis. Buildings with basements or timber, underpasses, crops or population vulnerable to flooding will have more risk than general targets with the same hazard. Considering not only the magnitude of the target but also the condition is necessary to establish a more effective climate change adaptation strategy. Since high-resolution spatial information and a lot of condition information are required in order to consider the condition of the target, a specific area will be selected as a case study and will be studied in the future.

The goal of future works is to forecast changes of future flood risk through risk assessment under the SSP climate change scenarios according to the global warming degree in the Republic of Korea and to suggest countermeasures. Quantitative flood risk will be assessed under the carbon neutrality or 1.5 °C global warming scenario through greenhouse gas reduction efforts. Regions where residual risks still exist despite achieving carbon neutrality will be detected, and the remaining risks of these areas quantitatively calculated. In addition, it will be analyzed how much the flood risk increases by region under the global warming scenario of 2.0 °C or higher, and how much adaptation effort will be needed to adapt to climate change will be studied.

## 5. Conclusions

This study evaluated the flood risk in Republic of Korea considering RCP climate change scenarios by applying the concept of risk, as set out in the IPCC AR5 report, consisting of hazard, exposure, and vulnerability. Compared to previous studies that used simple rainfall patterns as indicators, such as annual maximum rainfall and number of days with more than 110 mm of daily rainfall, this study utilized the design rainfall, which represents the local flood protection capability, as a flood threshold, in order to construct hazard indicators. High-resolution spatial images were constructed from data on buildings, roads, agriculture areas, and populations with the most flood damage in Republic of Korea, and the exposure was calculated by analyzing facilities and populations in low-lying areas. Furthermore, environmental and anthropogenic conditions that can directly increase or decrease river flooding and urban flooding were set as indicators and utilized as proxy variables.

As a result of risk assessment during the historical period, it was found that there were 43 areas with a higher severity of risk (19%) out of 229 administrative districts; 42 in the projected period under RCP 4.5 (18%); and 51 in the projected period under RCP 8.5 (22%). In the historical and projected periods under the RCP 4.5 scenario, the number of risk areas was similar to each other; however, in the projected period under the RCP 8.5 scenario, it was found that there were somewhat more risk areas than in the historical period. The risks in the historical period in the northern region were expected to gradually decrease, and the risks in the projected period in the central and southern regions were expected to increase. Sixteen administrative districts with the severity rating of "very high" were selected as the districts that most urgently needed to establish climate change adaptation measures. As a result of the indicator analysis of 16 districts, it was found that the hazard composite indicator had a huge impact on greater flood risk in 10 regions, the exposure indicator in 4 regions, the vulnerability indicator in one region, and the exposure and vulnerability composite indicator in one region. Most of the 16 districts did not experience a rainfall event exceeding the design rainfall of river and urban areas during the historical period; the rainfall in the districts with such an event was 1.17 times the design rainfall, evidencing that the excess rainfall was not high. In the projected period under the RCP 8.5 scenario, events exceeding the design rainfall were an average of two, and a maximum of four; the ratios of maximum rainfall and design rainfall were an average of 1.41, and a maximum of 2.95. In other words, the amendment of design rainfall standards, and the improvement of flood protection structures in river and urban areas, are required.

It is expected that the central government will be able to make decisions to enact the Climate Change Adaptation Plan by determining the priority of flood risk areas in each administrative district and analyzing the causes of flooding through the risk as-sessment method and results of this study. The local governments in the administra-tive districts with a high priority for flood risk works can use the results of this study to select a priority area for the installation of flood reduction facilities through the analy-sis of risk areas using high-resolution geospatial data.

**Author Contributions:** Conceptualization, I.Y.; Data curation, H.J.; Formal analysis, I.Y.; Funding acquisition, H.J.; Methodology, I.Y.; Project administration, H.J.; Resources, H.J.; Software, I.Y.; Supervision, H.J.; Visualization, I.Y.; Writing–original draft, I.Y.; Writing–review & editing, H.J. All authors have read and agreed to the published version of the manuscript.

**Funding:** This paper is based on the results of the research work "Assessment of climate change vulnerability and support tool" (2022-001-05), conducted by the Korea Environment Institute (KEI) upon the request of the Korea Ministry of Environment.

**Conflicts of Interest:** The authors declare no conflict of interest.

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
