# Peer review of "Flood Risk Assessment to Enable Improved Decision-Making for Climate Change Adaptation Strategies by Central and Local Governments"

_sustainability, doi:10.3390/su142114335_

Round 1

Reviewer 1 Report

This article examines the flood risk so as to improve decision-making for the adaption of cities to climate change. The topic is interesting and the article is well-written and easy to follow. Moreover, the scientific soundness is high. However, some issues that need to be addressed by the authors are the following:

Line 30. It is important to highlight that critical infrastructure exposure to flood zones is an important task in flood management plans. Despite the improvements in flood mitigation measures and technological advancements, floods continue to endanger human lives and infrastructures. This is mainly due to the increasing human settlements and economic assets in floodplains, land-use change, and climate crisis. The spatial knowledge of elements at risk provides useful information to governments for policy making (Stefanidis et al; 2022)

Stefanidis, S., Alexandridis, V., & Theodoridou, T. (2022). Flood Exposure of Residential Areas and Infrastructure in Greece. Hydrology, 9(8), 145.

Van Ginkel, K. C., Dottori, F., Alfieri, L., Feyen, L., & Koks, E. E. (2021). Flood risk assessment of the European road network. Natural Hazards and Earth System Sciences, 21(3), 1011-1027.

Qiang, Y. (2019). Flood exposure of critical infrastructures in the United States. International Journal of Disaster Risk Reduction, 39, 101240.

Line 40. Add some results from the latest report of the Intergovernmental Panel on Climate Change (IPCC) and provide future predictions for extreme events and flooding.

IPCC. Climate Change 2021: The physical science basis. In Contribution of Working Group, I to the Sixth Assessment Report of the Intergovernmental Panel on Climate Change; Cambridge University Press: Cambridge, UK, 2021.

Improve the legend in the figure 3.

It is preferable to separate the chapter Results and discussion.

Add some more explanation about the RCP emission scenarios.

Any goals for future research?

Author Response

Thank you for your careful review of our manuscript.

<Response to Reviewer 1 Comments>

This article examines the flood risk so as to improve decision-making for the adaption of cities to climate change. The topic is interesting and the article is well-written and easy to follow. Moreover, the scientific soundness is high. However, some issues that need to be addressed by the authors are the following:

  1. Line 30. It is important to highlight that critical infrastructure exposure to flood zones is an important task in flood management plans. Despite the improvements in flood mitigation measures and technological advancements, floods continue to endanger human lives and infrastructures. This is mainly due to the increasing human settlements and economic assets in floodplains, land-use change, and climate crisis. The spatial knowledge of elements at risk provides useful information to governments for policy making (Stefanidis et al; 2022)

Stefanidis, S., Alexandridis, V., & Theodoridou, T. (2022). Flood Exposure of Residential Areas and Infrastructure in Greece. Hydrology, 9(8), 145.

Van Ginkel, K. C., Dottori, F., Alfieri, L., Feyen, L., & Koks, E. E. (2021). Flood risk assessment of the European road network. Natural Hazards and Earth System Sciences, 21(3), 1011-1027.

Qiang, Y. (2019). Flood exposure of critical infrastructures in the United States. International Journal of Disaster Risk Reduction, 39, 101240.

(A) It was highlighted in the 1. Introduction chapter that flood exposure of critical infrastructure, human settlements, and economic assets is important for flood mitigation and management referring to the literature suggested by the reviewer.

  1. Line 40. Add some results from the latest report of the Intergovernmental Panel on Climate Change (IPCC) and provide future predictions for extreme events and flooding.

IPCC. Climate Change 2021: The physical science basis. In Contribution of Working Group, I to the Sixth Assessment Report of the Intergovernmental Panel on Climate Change; Cambridge University Press: Cambridge, UK, 2021.

(A) Projected flood risk results from the recently published Sixth Assessment Report of the Intergovernmental Panel on Climate Change are described in the introduction.

  1. Improve the legend in the figure 3.

(A) The legend of all maps, not just Figure 3, has been modified. We have improved the readability of the legend by modifying the legend's name, size, direction, and scale.

  1. It is preferable to separate the chapter Results and discussion.

(A) Results and discussions chapter have been separated from each other

  1. Add some more explanation about the RCP emission scenarios.

(A) Descriptions of RCP emission scenarios cited from the Fifth assessment reportof IPCC are described in the 1. Introduction chapter.

  1. Any goals for future research?

(A) The goal of future research is as follows and described in chapter 3.4.

The goal of Future works is to forecast changes of future flood risk through risk assessment under the SSP climate change scenarios according to the global warming degree in Republic of Korea and suggest countermeasures. Quantitative flood risk will be assessed under the carbon neutrality or 1.5°C global warming scenario through greenhouse gas reduction efforts. Regions where residual risks still exist despite achieving carbon neutrality are detected, and the remaining risks of these areas are quantitatively calculated. In addition, it is analyzed how much the flood risk increases by region in the global warming scenario of 2.0℃ or higher, and how much adaptation effort is needed to adapt to climate change is studied.

(Please see the attachment)

Reviewer 2 Report

Dear authors, I attached my corrections in the PDF.

All the best.

Author Response

Thank you for your careful review of our manuscript.

<Response to Reviewer 2 Comments>

L160-164 :

  1. This kind of assessments have been tested and proved at national scales before. Check and add the following examples:

A flood susceptibility model at the national scale based on multicriteria analysis Landslide risk index map at the municipal scale for Costa Rica. Flood risk index development at the municipal level in Costa Rica: A methodological framework. A comprehensive approach to understanding flood risk drivers at the municipal level.

(A) The introduction has been modified in consideration of the references recommended by the reviewer.

L392:

  1. Please try to first relate, contrast and discuss your results with Korea, after other similar countries, and finally with the whole world, trying to insert examples from all the continents. You are not posing your manuscript in an international framework.

(A) This study was compared with studies of domestic and foreign risk assessment and described in chapter 4.3 comparison with previous studies.

  1. Please separate your Discussion sections as the ones you used in Results and are coming from Methods

(A) Results and discussions chapter have been separated from each other

Reviewer 3 Report

This manuscript develops a flood risk index for Korea and applies projections of two climate change scenarios for its evaluation. This work will undoubtedly be useful for decision making by risk and water managers. I have only a few comments, but I would like all of them to be solved.

Introduction: The introduction is very repetitive and does not get to the point of the document without some twists and turns. It should consist of: (1) the general topic and justification of an argument/controversy, (2) the specific topic and its definition, (3) the state of the art on the specific topic, (4) the objective of the present study.

L46-52: I disagree, the terms risk and vulnerability are well established in the literature. I could leave dozens of references that clearly establish the same differences, but I will leave three references that in my opinion are the clearest.

https://doi.org/10.1007/978-1-4020-4200-3_13

https://doi.org/10.1080/02508060508691837

https://doi.org/10.1108/17595901211245189

L59 and L78: This reference would be useful here: https://doi.org/10.1016/j.envsci.2022.03.012

L64: It would be better to use the English version of the denomination : German Society for International Cooperation

L90-91: is repeated in Line 43-44

M&M

L114-115: Connect these two sentences.

L117-118: This was already established long before the IPCC report, see the reference by Kron et al (2005).

L118-120: See https://doi.org/10.1088/1748-9326/ac7ed9

L110-139: What is the point of defining all the different types of risk? This paper does not analyze all these types of risk, so there is no reason for them to be defined.

L140-148: This is what the paper analyzes, and these definitions are presented in the references I gave at the beginning.

Equation 2 and 4. It is not clear how they obtain the min and max values, in which time periods or the calculation.

Table 2. Although the indicators were chosen in discussion with experts (which is very valuable), I am surprised that only few indicators were chosen when more could have been included. See some examples: https://doi.org/10.1080/02626667.2011.583249

·        See others work in Korea for comparison and discussion: https://doi.org/10.3741/JKWRA.2013.46.1.35

·        Lim, K. S., Choi, S. J., Lee, D. R., & Moon, J. W. (2010). Development of flood risk index using causal relationships of flood indicators. KSCE Journal of Civil and Environmental Engineering Research, 30(1B), 61-70.

In figure 10 change the color of the “Risky population 200-300” because is very similar to the “risky agriculture area”.

Author Response

Thank you for your careful review of our manuscript.

<Response to Reviewer 3 Comments>

This manuscript develops a flood risk index for Korea and applies projections of two climate change scenarios for its evaluation. This work will undoubtedly be useful for decision making by risk and water managers. I have only a few comments, but I would like all of them to be solved.

(1) Introduction: The introduction is very repetitive and does not get to the point of the document without some twists and turns. It should consist of: (1) the general topic and justification of an argument/controversy, (2) the specific topic and its definition, (3) the state of the art on the specific topic, (4) the objective of the present study.

(A) Thank you for your kind comments to fix the introduction. The structure of the introduction has been modified to reflect the reviewer's opinion.

(2) L46-52: I disagree, the terms risk and vulnerability are well established in the literature. I could leave dozens of references that clearly establish the same differences, but I will leave three references that in my opinion are the clearest.

https://doi.org/10.1007/978-1-4020-4200-3_13

https://doi.org/10.1080/02508060508691837

https://doi.org/10.1108/17595901211245189

(A) We fully agree with the reviewer's comment that the definitions and differences of risk and vulnerability are clearly established. We should have mentioned that the methods of assessing risk and vulnerability differ depending on the researcher and the purpose of the study. This is because risk can be assessed in a variety of ways, such as indicators or numerical modeling. The sentences have been modified to reflect this.

(3) L59 and L78: This reference would be useful here: https://doi.org/10.1016/j.envsci.2022.03.012

(A) Thank you for recommending a suitable reference for this paper. Reference has been added.

(4) L64: It would be better to use the English version of the denomination : German Society for International Cooperation

(A) The full name of GIZ has been modified to the English version.

(5) L90-91: is repeated in Line 43-44

(A) The sentence mentioned by the reviewer has been deleted as it is deemed unnecessary.

M&M

(6) L114-115: Connect these two sentences.

(A) The two sentences are linked into one sentence.

(7) L117-118: This was already established long before the IPCC report, see the reference by Kron et al (2005).

(A) The concept of vulnerability was introduced in the IPCC 4th assessment report, and the concept of risk appeared in the 5th assessment report. The concept of risk was established a long time ago, but the 5th Assessment Report, a representative research report on climate change risks, was cited.

(8) L118-120: See https://doi.org/10.1088/1748-9326/ac7ed9

(A) A definition for multiform flood has been added to the manuscript by referring to literature recommended by reviewers.

(9) L110-139: What is the point of defining all the different types of risk? This paper does not analyze all these types of risk, so there is no reason for them to be defined.

(A) In order to decide a method suitable for this study among various risk assessment methodologies, the concept of risk applied in previous studies was introduced. In this study, the methodology that integrated the index-based risk assessment and the flood impact assessment through modeling was applied, and risks in various studies were introduced to show the applicability of this methodology.

(10) L140-148: This is what the paper analyzes, and these definitions are presented in the references I gave at the beginning.

(A) As the reviewer mentioned, this study did not define the concept of risk, but the previously defined concept of risk was used in this study. L140-148 was modified to reflect this.

(11) Equation 2 and 4. It is not clear how they obtain the min and max values, in which time periods or the calculation.

(A) The lowest risk or hazard during historical period among 229 administrative districts becomes the min value, and the highest risk or hazard becomes the max value. This explanation has been added in the section 2.2 Methods of assessing flood risk

(12 )Table 2. Although the indicators were chosen in discussion with experts (which is very valuable), I am surprised that only few indicators were chosen when more could have been included. See some examples: https://doi.org/10.1080/02626667.2011.583249

  • See others work in Korea for comparison and discussion: https://doi.org/10.3741/JKWRA.2013.46.1.35
  • Lim, K. S., Choi, S. J., Lee, D. R., & Moon, J. W. (2010). Development of flood risk index using causal relationships of flood indicators. KSCE Journal of Civil and Environmental Engineering Research, 30(1B), 61-70.

(A) Initially, a large number of indicators of hazards, exposure, and vulnerability were selected. After discussions with various experts (water resources, atmosphere science, etc.) with more than 10 years of experience, 23 indicators were recommended as shown in Table 2. Finally, 13 key indicators were selected, and the reasons why the other indicators were not selected are described in Sections 2.5.1-2.5.3. In this study, a flood map based on numerical modeling was used to calculate exposure indicators. The parameters for simulating flooding include various spatial information such as flow direction, slope, channel width, roughness, and impervious etc. calculated through digital elevation model, soil map, and land use map. Since regional topographic information was considered using the flood map, indicators related to digital elevation model, soil map, and land use map were excluded.

(13) In figure 10 change the color of the “Risky population 200-300” because is very similar to the “risky agriculture area”.

(A) The color of the risky population has been modified in Figure 10

Round 2

Reviewer 1 Report

The authors addressed all the reviewer comments